



# Boundary layer evolution over the central Himalayas from Radio Wind Profiler and Model Simulations

Narendra Singh[1*], Raman Solanki[1, 2], Narendra Ojha[3], Ruud H. H. Janssen[3], Andrea Pozzer[3] and Surendra K. Dhaka[4]

[1]Aryabhatta Research Institute of Observational Sciences, Nainital, India
[2]Department of Physics and Astrophysics, University of Delhi, India
[3]Department of Atmospheric Chemistry, Max Planck Institute for Chemistry, Mainz, Germany
[4]Radio and Atmospheric Physics Lab., Rajdhani College, University of Delhi, India

*Correspondence to*: Narendra Singh (narendra@aries.res.in)

**Abstract.** We investigate the time-evolution of the Local Boundary Layer (LBL) for the first time over a mountain ridge at Nainital (79.5°E, 29.4°N, 1958 m amsl) in the central Himalayan region, using a Radar Wind Profiler (RWP) during November 2011 to March 2012, as a part of the Ganges Valley Aerosol Experiment (GVAX). We restrict our analysis to clear-sunny days, resulting in a total of 78 days of observations. The standard criterion of the peak in the signal-to-noise ratio (SNR) profile was found to be inadequate in the characterization of Mixed Layer (ML) top at this site. Therefore, we implemented a criterion of SNR > 6 dB for the characterization of the ML and the resulting estimations are shown to be in agreement with radiosonde measurements over this site. The daytime average observed boundary layer height ranges from 440±197 m in November (late autumn) to 766±317 m in March (early spring). The observations revealed a pronounced impact of mountain-topography on the LBL dynamics during March, when strong winds (> 5.6 m s⁻¹) lead to LBL heights of 650 m during nighttime. The measurements are further utilized to evaluate simulations from the Weather Research and Forecasting (WRF) model. WRF simulations captured the day-to-day variations up to an extent ($r^2 = 0.5$), as well as the mean diurnal variations (within 1-sigma variability). The mean biases in the daytime average LBL height vary from -7% (January) to +30% (February) between model and observations, except during March (+76%). Sensitivity simulations using a Mixed Layer model (MXL/MESSy) indicated that the springtime overestimation of LBL would lead to a minor uncertainty in simulated surface ozone concentrations. However, it would lead to a significant overestimation of the dilution of black carbon aerosols at this site. Our work fills a gap in observations of local boundary layer over this complex terrain in the Himalayas, and highlights the need for yearlong simultaneous measurements of boundary layer dynamics and air quality to better understand the role of lower tropospheric dynamics in pollution transport.





**Keywords:** Local boundary layer, Radar Wind Profiler, central Himalayas, WRF, GVAX

# 1 Introduction

Measurements on the diurnal evolution of the planetary boundary layer (PBL), the marine boundary layer (MBL), and in particular over high altitude complex terrains the local boundary layer (LBL), are essential to understand the vertical transfer of momentum, energy and the mixing of pollutants in the lower troposphere. The depth or height of the PBL is a fundamental parameter in numerical simulations of regional meteorology and air quality. The height of the mixed layer (ML) is a measure of the effectiveness of the energy transfer from the sun to the earth's surface and thereby to the lower atmosphere and is therefore important for understanding various atmospheric processes (Stull, 1989; Garratt, 1993). Despite being simple to understand conceptually, the measurement of ML height is rather difficult (Coulter and Holdridge, 1998). The most traditional method is the analysis of potential temperature and specific humidity profiles, which are obtained from radiosonde ascents (Hooper and Eloranta, 1986). However, in the past three decades several remote sensing devices such as the Light Detection and Ranging (LiDAR), Sound Detection and Ranging (SoDAR), Radio Acoustic Sounding System (RASS), GPS occultation measurements (Basha and Ratnam, 2009; Xie et al., 2012) and Radio Wind Profilers (RWP) have become indispensable tools to probe troposphere through atmospheric boundary layer. A review and detailed studies carried out earlier on PBL evolution over various locations are given elsewhere (Friehe, 1987; Garratt and Taylor, 1996).

It is essential to mention that most of the studies cited here and the references therein, are confined to mid and high-latitude regions, and that very few efforts have been made to characterize the PBL evolution over the Indian subcontinent and the complex terrain of the Himalayan region. The available studies, over the Indian region have described the role of the boundary layer in trapping down and transporting pollutants upward to pristine high altitude locations and to other continental locations using satellite and ground based measurements, are given subsequently.

The variations in boundary layer height are suggested to influence the dispersion of air pollutants near the earth surface (Guatam et al., 2007). Deeper boundary layers could also mix residual layer air with higher ozone concentrations with the air mass near the surface (Reddy et al., 2012). The investigation of local boundary layer evolution and associated mixing is additionally important over the pristine Himalayas just north to the densely populated and polluted Indo-Gangetic Plain (IGP) to understand the role of regional pollution on the air quality above Himalayas (Sarangi et al., 2014). In previous studies, the potential influence of distantly transported and IGP aerosol emissions on air quality in the central Himalayas has been demonstrated using LIDAR measurements (Solanki et al., 2013; Solanki and Singh, 2014), as well as the convective mixing of photochemical pollution (Ojha et al., 2012). Continuous measurements of LBL evolution and mixing have not been available until a recent intensive field campaign "Ganges Valley Aerosol Experiment (GVAX)" (Kotamarthi, 2010; Kotamarthi and Satheesh, 2011; Manohanarn et al., 2014). However, systematic and high resolution measurements of the PBL evolution over the IGP region and the LBL evolution over the Himalayan region for a complete seasonal cycle are still missing.



During the GVAX campaign at ARIES, Nainital regular radiosonde launches (comprising of four launches per day), were made, but a clear picture of LBL or ML evolution was not possible, since four radiosonde profiles a day provided only snapshots of the atmosphere. On the other hand, for the first time, RWP was operated for wind measurements; which also provides continuous measurements of boundary layer (Angevine et al., 1994; Coulter and Holdridge, 1998) with a finer

temporal resolution (15 minutes). Therefore, the RWP is the best possible tool to determine ML evolution over the site. The RWP also gives a deep insight into the vertical structure of the boundary layer in general, which is crucial in understanding dynamic meteorology, pollutant transport and dispersion (Xie et al., 2012). Studies on ABL dynamics over complex terrain have been made by Reddy et al. (2006) and Kalapureddy et al. (2007) over a moderately hilly terrain in Gadanki valley region in Southern India. However, no such studies have been made over the Himalayas for the highly contrasting winter and

spring seasons. The importance of this study can be understood by the fact that a strong diurnal cycle can appear in the LBL or ML over hilly terrain, under the fair-weather conditions which are suitable for boundary layer (Reddy et al., 2002).

The main objectives of our study are:

(1)     To investigate the diurnal, and monthly variations in LBL evolution over the central Himalayas using RWP observations.

(2)     To evaluate boundary layer height, as simulated by a regional model (WRF), using a setup similar to the one evaluated and extensively used over the Indian region (e.g. Kumar et al., 2012a; Kumar et al., 2012b; Sarangi et al., 2014; Ojha et al., 2015).

(3)     To assess the influence of model uncertainties in boundary layer height on air quality simulations, by conducting sensitivity runs with a mixed layer model (MXL/MESSy).

In this study, we implemented a new criterion for the SNR that captures nearly systematic  feature of LBL evolution with time, since over a mountain peak, the structure of the LBL is not as prominent as that as in high pressure regions or over flat terrains.   The estimation method using the RWP has been validated with ML heights from radiosonde profiles. We investigate the mean diurnal variations of ML height during months covering the late autumn (November), winter (Dec-Jan-Feb) and early spring (March). The impact of mountain topography on LBL evolution in the presence of strong winds during

nighttime is also investigated. Finally, we compare the day-to-day and diurnal variations in LBL height deduced from RWP measurements with simulations from a regional model (WRF). The implications of model-observation biases on air quality studies are also discussed.

The manuscript begins with a brief description of the observation site, the RWP instrumentation and the mixed layer determination methodology in the Section 2. The Weather Research and Forecasting (WRF) model that is employed to

simulate the temporal variations in boundary layer height and the Mixed layer model (MXL/MESSy) that is used for sensitivity simulations of air quality are described in Section 3. The results and discussion are presented in Section 4, followed by the summary and key conclusions in Section 5.





## 2 Observational site and Methodology

### 2.1 Observational site

The observational site is located at a mountain top called Manora Peak (79.5°E, 29.4°N and 1958 m amsl), near Nainital, a high altitude station in the central Himalayas. The observational site has 600 meters deep valleys on the eastern and western

sides and the valley axis are aligned approximately in the SE and SW direction, respectively. To the north of the peak the topography is very craggy and rising; towards the south, the mountains are gradually sloping into the adjacent plains of Haldwani. The mountain peak has a slope of approximately 25 degrees on the eastern and western side, and is covered with a dense forest on all sides, consisting mainly of trees and shrubs. All measurements presented in this study were made on the mountain peak with no obstructions such as buildings or trees in any direction. The site (Sagar et al., 2004) has been subject

of intensive studies on synoptic wind patterns and trace gases (Sarangi et al., 2014), balloon soundings (Ojha et al., 2014) and aerosol extinction profiles (Solanki and Singh, 2014). A detailed overview of the measurement site and the atmospheric science research conducted there has been provided in a recent review paper (Sagar et al., 2015).

### 2.2 RWP Instrumentation and Mixed Layer Determination

The 1290 MHz radar wind profiler (RWP), developed by DeTect, Inc., measures wind profiles and the backscattered signal

between 0.1 and 6 km. The system is capable of changing the beam pointing angle on a pulse-by-pulse basis. The profiler collected a sample every 30 s, completing a cycle of all five beam positions every 150 s. By selecting a beam elevation of 90 degrees, SNR profiles from a vertical beam position only were incorporated in this analysis. The SNR data for the vertical beam are available with a resolution of 62.6 m, with the first range bin at 124 m.

The ML height is inferred from the signal-to-noise ratio (SNR) recorded by the RWP. The underlying theory has been

discussed by White et al. (1991a, b). The principal source of scattered signal to RWP is moisture, which is the primary constituent of inhomogeneity in the radio refractive index of the air. These wind profilers basically detect the fluctuations in radio refractive index caused by vertical humidity and temperature gradients. These inhomogeneities are characterized by the refractive index structure parameter $C_n^2$ (Tatarskii, 1971; Green et al., 1979; Gossard et al., 1982; Gage, 1990; Raghavan, 2003; Singh et al., 2009). The profiler SNR at a given range is directly proportional to $C_n^2$ (Ottersten, 1969).

The vertical structure of the PBL (with height $Z_i$) consists of three different layers. The first layer is the surface layer starting from ground to $0.1(Z_i)$, the second layer is the ML extending from $0.1Z_i$ to $0.8Z_i$ and the third layer is the entrainment zone from $0.8(Z_i)$ to $1.2(Z_i)$. Above the PBL is the free troposphere. Generally, over a plain site the top of the ML is characterized by increased turbulence, and strong temperature and moisture gradients, which are represented by a peak in the SNR profile, marking the top of the ML (Simpson et al., 2007). However, this study is carried out over a high altitude site (2 km amsl) and

because of the lack of prominent features over the mountainous site the general methods of ML height determination over plains cannot be directly applied. The complex mountainous topography generates its own LBL (e.g. Kossmann et al., 1998), which is different from the PBL in some aspects. One of those is that the SNR profile, instead of exhibiting a peak in the





profile, shows a gradual decrease with altitude, as depicted in Fig. 1.One possible explanation for this aspect of the LBL could be the difference in the characteristics of entrainment zone over plains and mountainous terrain, since the entrainment zones basically form between the tops of highest thermal plumes and deepest part of the sinking dry air penetrating through the free troposphere. In contrast to a flat terrain, mountainous or complex terrain may not have that prominent formation of the entrainment zones due to complex topography, as convective mixing over the ridge may be influenced by slope winds and occasional strong horizontal flows. Therefore, in this study we have considered the region of SNR > 6 decibels (dB) as the LBL over the site and validated this method by comparing the results with those deduced from the vertical profiles of potential temperature and specific humidity from radiosonde (Vaisala RS92-SGP) ascents. In case of RWP SNR profiles showing multiple transitions across the 6 dB threshold, the maximum altitude was selected as the ML height, as shown by the 1000 UTC profile in Fig. 1. Although averaging time of 30 minutes is considered for parameters determining the evolution of PBL, but for the LBL study over the site we have considered the averaging time of 15 minutes (for mean SNR profiles) for better statistical average and taking into the mixing time scale, since the LBL is in a much more dynamical state i.e considering the convective mixing and the topographic effects, as compared to PBL. The contour plots of SNR were also analyzed visually in order to confirm the reliability of the ML height estimations at the 6 dB threshold. However, the SNR is proportional to refractive index structure parameters, hence can further be subjected to derive reflectivity and the turbulence parameters that again are the tracers of LBL or PBL evolution.

The vertical profiles of meteorological parameters obtained from the radiosonde ascents are also utilized to provide the vertical structure of the atmosphere for comparison with RWP measurements. The hourly wind speed observations made through a collocated automatic weather station (AWS) over the site are also incorporated to understand the state of the surface layer over the site.

## 3 Model Simulations

### 3.1 Weather Research and Forecasting (WRF) model

This study uses the version 3.5.1 of the Weather Research and Forecasting (WRF) model to simulate the temporal variations of boundary layer height at Nainital during the study period. Meteorological fields from NCEP Final Analysis (FNL) data available at the spatial resolution of 1 degree and temporal resolution of 6hours has been used to provide the initial conditions and the lateral boundary conditions in the model. Simulations were performed for two different spatial resolutions. The simulations from the coarser domain (15 km x 15 km) were used to provide the initial and boundary conditions for the higher resolution domain (5 km x 5 km). Both model domains and the topography of this region are shown in Fig. 2. The errors in the simulated meteorology in the coarser domain were limited by nudging of the temperature, water vapor and horizontal winds with a nudging coefficient of 6 x $10^{-4}$ per second at all the vertical levels (Kumar et al., 2012; Ojha et al., 2015).



The cloud microphysics was represented by the Thompson microphysics scheme (Thompson et al., 2008). The longwave radiation has been calculated using the Rapid Radiative Transfer Model (RRTM) (Mlawerwt al., 1997) and shortwave radiation is calculated using the Goddard shortwave scheme (Chou and Suarez, 1994). The surface layer has been parameterized using the Monin-Obukhov scheme (Janjic, 1996). The Noah Land Surface Model, which utilizes the Unified

NCEP/NCAR/AFWA scheme with soil temperature and moisture in four layers, has been used to parameterize the land surface processes (Chen and Dudhiya, 2001). The planetary boundary layer dynamics were parameterized using the Eta operational Mellor-Yamada-Janjic (MYJ) scheme, which is based on a one dimensional prognostic Turbulent Kinetic Energy (TKE) scheme with local vertical mixing (Janjic, 2002). The cumulus parameterization was based on the new Grell (G3) scheme for the coarser domain, while it has been turned off for the nested domain as shown in the Supplementary Material

(Fig. S1). Detailed discussions on simulations using the WRF model, the chosen physics options and meteorological nudging, including evaluations over the Indian region, can be found in recent studies (Kumar et al., 2012a; Kumar et al., 2012b; Sarangi et al., 2014; Ojha et al., 2015). Hourly model output from the nested 5 km x 5 km simulation has been used in the analysis.

**3.2 MiXed Layer model (MXL/MESSy)**

The implications of biases in the WRF simulated boundary layer dynamics, as compared to RWP measurements, on air quality simulations have been investigated by conducting sensitivity simulations with the MiXed Layer/ Modular Earth Submodel System model (MXL/MESSy, version 1.0; Janssen and Pozzer, 2015). Although the BL dynamics are represented in a different way in MXL/MESSy than in WRF, we can use the former to obtain insight in the effects of an overestimation of BL height on modeled chemical species concentrations. MXL/MESSy has been developed recently as a column model in

the MESSy framework (Joeckel et al., 2010). Within MXL/MESSy, the MXL submodel accounts for the dynamics of the convective boundary layer during daytime, by explicitly calculating BL-free troposphere exchange of scalars and chemical species through entrainment. Through the coupling of MXL with other MESSy submodels for processes that are relevant for atmospheric chemistry, MXL/MESSy can be used to evaluate the influence of BL dynamics on atmospheric chemistry. The mixed layer theory states that under convective conditions, strong turbulent flow causes perfect mixing of quantities over the

entire depth of the ABL (Vilà-Guerau de Arellano et al., 2015). Therefore, scalars and reactants in the convective boundary layer are characterized by a well-mixed vertical profile over the whole depth of the ABL. In MXL, the transition between the well-mixed BL and the free troposphere is marked by an infinitesimally thin inversion layer. In this study, chemical transformations are represented by the Mainz Isoprene Mechanism 2 (MIM2; Taraborrelli et al., 2009). Black Carbon (BC) is treated as a passive tracer, which is appropriate at the short time scale (6 hours) of our simulations.



## 4 Results and discussion

### 4.1 Estimation of ML height

Figure 3 and Fig. 4 depict the diurnal variability (24 hour cycle) of the RWP measured SNR profiles (15 minutes averaged) on two contrasting representative days in different months (with Sensible heat flux of 17 and 50 W m$^{-2}$ during the December and March). For each day, data from four radiosonde launches (approximate launch time of 0000, 0600, 1200 and 1800 UTC) were also available, which are used to investigate the profiles of potential temperature and specific humidity for comparison with ML height estimated from the RWP. A very clear smooth diurnal variation in SNR is observed on December 17, 2011, as illustrated in Fig. 3; this feature is characteristic of a mountain LBL in the winter under calm wind conditions (wind speed < 2 m s$^{-1}$), the evolution is seen from 0200 to 1300 UTC i.e., 0730 h to 1830 h local time (LT= UTC+5.5 h), attaining a peak of 500 m at noon (0700 to 0800 UTC). During nighttime (1300 to 0200 UTC) the LBL is extremely shallow in depth, and under the RWP lower detection range (i.e., 124 m). The specific humidity and potential temperature profiles also show a clear demarcation between the LBL and the free troposphere during daytime (0622 UTC), and the height of the inversion (~300 m) agrees quite well with the RWP-derived LBL height at the same time. The other radiosonde profiles (2350, 1148 and 1741 UTC) show stable potential temperature profiles, which is consistent with a stable (nocturnal) boundary layer.

Figure 4 shows the ML height evolution on March 15, 2012, which is considered as representative of the early spring season over the site, based on the mean diurnal variability of the ML during March. Two distinct growth phases of the ML are observed during the 24 hour cycle, with one starting at 0500 UTC and the other at 1300 UTC. The growth and decay in ML from 0500 to 1200 UTC is consistent with the diurnal cycle of incoming solar radiation. The peak height of the ML is above 900 m, and attained from 0800 to 1000 UTC. A second growth phase in the ML depth starts from 1300 UTC, and the ML remains stable from 1500 to 2030 UTC with ML height reaching up to approximately 700 m. The minimum ML height of 300 m is observed from 0000 to 0330 UTC. The vertical profile of specific humidity shows a strong inversion between 400 to 500 m, which is consistent with the estimated ML height derived from the RWP at the respective radiosonde launching time (0608 and 1139 UTC). However, a clear inversion in the potential temperature profile is seen only for the 1139 UTC ascent and a very weak gradient is also seen for the 0608 UTC profile, for both the profiles the inversion occurs at approximately 500 m pointing at a convective ABL. It is also notable from 1139 and 1733 UTC radiosonde launches, that inversion in the potential temperatures and specific humidity are taking place between 1500m and 2000m which could be due to advected residual layers.





## 4.2 Diurnal Variations

The comparison of monthly average diurnal variations in the ML height during the period from November 2011 to March 2012 is presented in Fig. 5. The ML height was estimated only for the clear sunny days (without any large mesoscale activity), which were selected for each month through the sky condition log book that is maintained at the site and also by the visual inspection of sky camera videos. Table 1 shows the number of clear sky days for the different months.

A clearly defined diurnal variation in ML height is observed from November to February with least diurnal variability in November and December, exhibiting a peak value of 500 m from 0700 to 0900 UTC. From January to March, the mean ML height shows overall a gradual increase, with peak values above 800 m. In general, the nocturnal boundary layer height remains below 300 m from 1600 to 0100 UTC. Towards the start of spring in the month of March, the ML height shows a distinct behavior with a mixing depth as high as 650 m during 1400 to 2200 UTC, which is attributed to the strong horizontal flow hitting the mountain (discussed in section 4.3), which gives rise to the lifting motions, and hence the rise of a few 100 meters in the mixing depth.

## 4.3 Impact of mountain topography

The impact of mountain topography can be clearly noticed by comparing the diurnal variability of the ML height in December and March. Figure 6 shows the diurnal variability in wind speed during December 2011 and March 2012 measured by a collocated AWS at the site. In December, when the winds are calm ($< 2 - 3$ m/s), the LBL growth starts around 0300 UTC. From then onwards, the ML height increases gradually and approaches a maximum of about 700 m in the afternoon, which appears to be in phase with the intensity of sunshine. ML height drops after 1000 UTC, and in the evening from 1300 UTC onward stays unchanged during rest of the night (1500 to 0300 UTC). However, in March when the wind speeds are more than doubled, reaching values as high as $6 \pm 3$ m/s, the LBL settling in nighttime hours is hindered since such high winds cause significant wind shear, thereby generating turbulent eddies and increasing the vertical mixing of surface–layer air (Solanki et al., 2015), leading to a second growth and decay phase in ABL depth from 1300 to 2330 UTC. This second growth phase can also be understood as the deepening of surface layer, transforming into residual layer (Henne et al., 2014). A decrease in wind speed is not observed from 2300 to 0300 UTC when the ABL decreases in depth, this could be attributed to the cooling of the surface beyond a certain extent, leading to the formation of a thin stable nocturnal boundary layer decoupled from the residual layer above. High wind speeds near the surface are one of the characteristics of such stable layers.

## 4.4 Comparison with Model Simulations

In this section, we utilize the ML height observations from the RWP to evaluate the simulation from a regional model (WRF) that has been used in previous studies for simulating the meteorology and regional air quality over Indian region (e.g.





Kumar et al., 2012; Sarangi et al., 2014; Ojha et al., 2015). We focus on the capability of the model, with a setup similar to the one used in aforementioned studies, in capturing the diurnal and day-to-day variability in mixing depth. Errors in mixing depth could lead to considerable uncertainties in the dispersion and mixing of the air pollutants over this region. For comparison, the model output has been obtained for the same selected clear-sunny days as the RWP measurements. Since

WRF output is instantaneous hourly data, instantaneous hourly data has also been used from the RWP for comparison.

Figure 7a shows the comparison of daytime (0500-1000 UTC) average boundary layer height from RWP measurements and WRF simulations from November 2011 to March 2012. The day-to-day variations in the daytime boundary layer height, as observed from the RWP, are captured by the model during late-autumn and throughout winter; however, the model shows significantly higher boundary layers towards the start of the spring (March). Figure 7b shows a correlation analysis between

model simulations and observations of daytime boundary layer height. Overall the model and observations are in reasonable agreement during the study period ($r^2 = 0.5$). The monthly statistics of the model-observation comparison are given in the Table 2. The mean bias is the average difference (WRF - RWP) for the selected days of each month, and similarly the percentage bias is the percentage difference (WRF - RWP) normalized with respect to the mean RWP measurements.

On average, the noontime boundary layer height is slightly underestimated from November to January, by 47 to 85 m (7-

17%). The variability (1-sigma standard deviation) in the modeled boundary layer height is also lower (90-149 m) than in the observations (197- 289 m) during this period. However, during the transition from winter to spring (February), the variability and biases are observed to change. The model overestimates the boundary layer height by 204 m (30 %) and shows a much higher variability (439 m) than the observations (268 m), in contrast to the months of Nov-Jan. The overestimation of the boundary layer height is much more during March, as high as 584 m (by 76.2 %), and the variability in the modeled

boundary layer height is more than double (664 m) than that in the observations (317 m).

The mean diurnal variations in boundary layer height are compared between model and observations for all the months of the study period (Fig. 8). The model simulated diurnal variations are in agreement with the measurements, as average daytime as well as nighttime values are generally within 1-sigma variation of each other. An appreciable disagreement between model and measurements is only seen towards the evening hours (1000 to 1200 UTC), when the boundary layer

height shows a gradual decrease in the RWP measurements, whereas the model simulations exhibit a rapid decrease in the boundary layer height.  In contrast to the period from November to February, the boundary layer height is overestimated throughout the day in March, with noontime mixing layer depth about two times higher than in the observations (Table 2). We suggest that a mixing layer depth about two times higher than in the measurements could lead to significant dilution or lead to additional entrainment when used in air quality models. In the next subsection, we explore the possible implication of

model biases in March above this site on air quality simulations using a 1-D mixed layer model.



### 4.5 Effects of boundary layer height overestimation on air quality simulations

In this section, we use MXL/MESSy (Section 3.2) to investigate the influences of uncertainties in boundary layer height simulated by regional models on air quality simulations. For this purpose, we combined the effects of the boundary layer overestimation during March with the available information on vertical gradients and emissions at Nainital to assess the

impact on two chemical tracers: Ozone ($O_3$) and Black Carbon (BC).

First, we set up MXL/MESSy to reproduce the observations of boundary layer height, potential temperature, specific humidity and wind speed (Supplementary Material-Fig. S3) for the representative day of March 15, 2012. Heat fluxes were prescribed to the model using typical values for March, based on our observations at this site using sonic anemometer measurements (Solanki et al., 2015). Initial and boundary layer conditions are given in table 3. A set of simulations was

performed (Fig. 9) to identify the simulation that reproduces the observations best, which show a rapid boundary layer growth in the morning and the simultaneous increase of potential temperature and specific moisture. In addition to the surface fluxes of sensible and latent heat, which are the main drivers of boundary layer growth, large-scale subsidence and advection of cool and moist air were required to reproduce the observations. The assumption of large-scale subsidence in MXL/MESSy is consistent with the vertical downward wind speed of a few cm/s in the WRF simulations over the region.

Figure 9 shows the results from 4 simulations. The simulation which is closest to the observations (best run) includes both large-scale subsidence and advection of cool and moist air. The simulation without any subsidence leads to an overestimation (by ~ 250 m) of the maximum boundary layer height. The third simulation, in which there is no advection, overestimates LBL height and temperature and underestimates specific moisture. The final simulation, without advection and subsidence, yield an overestimation of the LBL height by 400 m.

The ozone gradient between the boundary layer and free troposphere is assumed to be 5 ppbv, as reported for the spring (MAM) season over this site based on ozonesonde observations (Ojha et al., 2014). Further, ozone-poor air is transported upwards from the valley during daytime (Ojha et al., 2012; Sarangi et al., 2014). MXL/MESSy consists of two boxes and cannot explicitly account for upslope flows. Therefore, we mimic its effect by assuming a constant ozone loss at the bottom of the lowest box. The effect of variations in the boundary layer height on $O_3$ concentrations is found to be relatively small

for the different simulations. The ozone mixing ratios differ by less than 2 ppb when the maximum boundary layer height is ~1350 m, as compared to the more realistic ~1000 m.  This is due to the relatively small gradient in ozone mixing ratios across the LBL-free troposphere interface, which makes sure that entrainment of air from the free troposphere only dilutes the LBL ozone concentrations by a small fraction.

Unfortunately vertical profiles of black carbon are not available over Nainital site, and therefore measurements at another

high altitude site Hanle (78.96$^{o}$E, 32.78$^{o}$N; ~ 4.5 km above sea level) in the Himalayan region were used to estimate the vertical gradient of BC. Average BC concentrations are reported to be 110 ng/m$^3$ at Hanle as compared to 1340 ng/m$^3$ for our site (2 km above sea level) during spring (Dumka et al., 2010; Babu et al., 2011). The measurements show that BC concentrations increase during the day, which is attributed to the upward mixing of air masses from the nearby valley in the





polluted Indo-Gangetic plain region (e. g. Dumka et al., 2010). This leads to a net increase in BC concentration during daytime, although entrainment of BC-poor free tropospheric air is also active. Emissions of BC were initially set at their values in the nearby valley, which were in the order of $10^{-12}$ kg m$^{-2}$ s$^{-1}$, based on HTAP inventory (Janssens-Maenhout et al., 2015). The emissions were then tuned to obtain the best comparison with the observations and we found that a sinusoidal

emission profile with a maximum of 5.5 x $10^{-12}$ kg m$^{-2}$ s$^{-1}$ gave the best results.

Modeled BC concentrations are more sensitive to the variations in boundary layer height than those of ozone, because of a larger vertical concentration gradient. BC concentrations are simulated to be lower by ~300 ng/m$^3$ in the case when the boundary layer height is ~ 350 m higher, due to missing effects of subsidence and advection. Note that with advection, we only mean advection of cool and moist air, and not advection of BC. The cool and moist air leads to a decrease of the BL

temperature and an increase of the humidity. Consequently, the potential temperature and specific humidity gradient between the BL and the free troposphere increase, and therefore entrainment decreases (Janssen et al., 2013). For the same BC emissions, this means that more BC is trapped in a shallower BL and that it is diluted less with BC-poor air from the free troposphere. Subsidence acts to oppress the BL growth, but enhances entrainment (Janssen et al., 2013). Therefore, the BC concentration is diluted a little less in the simulation without subsidence.

Besides subsidence and advection, the conditions in the free troposphere can affect the BL dynamics as well. Therefore, the effect of boundary layer variations on BC at Nainital are further explored by conducting simulations for different gradients of potential temperature from boundary layer to free troposphere and for different lapse rates of the potential temperature in the free troposphere (Fig. 10). Initial mixing ratio and surface emission fluxes are given in table 4. We find that when the initial potential temperature gradient ($\Delta\theta_0$) is increased from 0.2 to 2.0 K, the BL growth is suppressed in the first hours of

the simulation. Consequently, the BC emissions are concentrated in a shallower BL and therefore overestimated compared to the observations by 350 ng/m$^3$ (maximum). However, during the course of the day, the initial temperature barrier is overcome and the effect on the simulated BC concentration is reduced to 200 ng/m$^3$ at 09:00 UTC. Finally, the potential temperature lapse rate ($\gamma_\theta$) determines the growth rate of the BL: if it is small, the potential temperature difference between BL and free troposphere grows less with increasing BL height than when it is large. When $\gamma_\theta$ is set to 0.002 K/m instead of

0.0065 K/m, as in the control experiment, the BL growth is much stronger and the BL height reaches 1650 m at 09:00 UTC, which is an overestimation of about 700 m, and comparable to the overestimation by WRF. Consequently, BC concentrations are diluted much more and are underestimated by up to 700 ng/m$^3$. For a $\gamma_\theta$ of 0.009 K/m, the BL height is underestimated by about 150 m at maximum. The resulting error in the BC concentration compared to the control experiment is 300 ng/m$^3$.

Our analysis suggests that an overestimation of boundary layer height has a minor effect on ozone concentrations (of less than 2 ppb), but a significant effect on BC concentrations (of ~300 up to 700 ng/m$^3$). The effect of an overestimation of BL height on the concentration of a species is highly dependent on the vertical gradients of the species above our site. Ozone has almost equal concentrations in the BL and free troposphere above Nainital, but black carbon has sharp gradients above 2 km. We suggest that effects of boundary layer dynamics could be much higher in the nearby Gangetic basin, where the gradients





will be much steeper due to strong surface sources and intense local photochemistry. More simultaneous measurements of boundary layer dynamics and trace species are highly desirable in the northern Indian region to understand the extent up to which boundary layer dynamics influences the air quality.

## 5 Summary and Conclusions

We presented the continuous measurements of ML height over a mountain peak in the central Himalayas from November 2011 to March 2012, obtained through state of the art instrumentation employed as a part of the GVAX intensive field campaign. RWP measurements of SNR are utilized for the first time over the central Himalayas to estimate mixing and boundary layer height and covered the mixing depth variations during late-autumn, complete winter and early spring. The criterion of SNR > 6 dB for identifying ML depth was found to be adequate, and yielded a reasonable comparison boundary

layer height derived from the inversion in potential temperature profiles obtained from radiosonde launches. The results show that the LBL over the site undergoes clear diurnal variations in all months from November to February, attaining peak heights between 0700 to 0800 UTC and remaining stable during the night, having minimum height (1800 to 0100 UTC). However, in the month of March the LBL continues to decrease in depth till 1330 UTC (falling up to 500 m) and afterwards rises again and remains stable at 650 m from 1400 to 2200 UTC. The strikingly larger depth of the LBL can be attributed to

the strong winds over the site during the night, which results in strong orographic lifting over the site. The study re-establishes the fact that RWP gives the better temporal estimation of ML heights compared to balloon borne and other such measurements. As RWP provides the volume scattering from the turbulent scales to which radar is sensitive and receives backscatter power over a larger aperture that provides insight into the LBL dynamics through continuous measurements with fine vertical resolution.

The observations are further utilized to evaluate high resolution simulations from a regional model (WRF). WRF-simulated day-to-day variations in the noontime boundary layer height were in reasonable agreement with the RWP observations ($r^2$ = 0.5). Additionally, the monthly average diurnal variations in boundary layer height from model and observations are generally within the 1-standard deviation variability. The mean biases in the daytime boundary layer height are estimated to be -7 to +30 % from November to February, but a large overestimation of ~76 % was seen towards the early spring (March).

Our study fills a gap by providing a continuous observational dataset on the boundary layer dynamics over a geographically complex and environmentally important region of the central Himalayas. We found that while the regional model simulates the boundary layer evolution well during the post-monsoon season and winter, it shows large biases towards spring. This highlights the need to extend boundary layer observations to entire seasons of spring and summer.

Sensitivity simulations using MXL/MESSy were conducted to assess the impact of the uncertainty in ML height on air

quality simulations, thereby providing insight in the influence of subsidence and advection processes on boundary layer dynamics over the site. We analyzed effects on two chemical tracers: ozone and black carbon. We find a relatively small effect (~2 ppb) of overestimated boundary layer height during March on surface ozone concentrations at Nainital. In



contrast, significant dilution was found in case of black carbon (by 300-700 ng/m$^3$), due to the overestimation of boundary layer height.

It should be noted that spring is a period of strong winds near the surface, maximum solar radiation and the highest pollution loading over this region, which is followed by torrential rains of the monsoon season. Since regional photochemistry and

5  convective mixing are intensified during spring over the Gangetic basin, we suggest that year long measurements with instrumentation such as RWP are highly desirable to understand the influence of boundary layer dynamics on the mixing of pollution.

**Acknowledgements**

The RWP, radiosonde and surface observations were carried out as a part of GVAX campaign in joint collaboration among

10  Atmospheric Radiation Measurement (ARM), Department of Energy (US), Indian institute of Science (IISC) and Indian Space Research Organization (ISRO), India. We are thankful to R. L. Coulter for taking care of technical aspects of the RWP measurements. Use of NCEP FNL reanalysis data as input to WRF model is acknowledged. We thank Director, ARIES for providing the necessary support. Mr. Raman Solanki is thankful to the Indian Space Research Organization for sponsoring fellowship for his Ph.D. research work under ABLN&C: NOBLE project. WRF simulations were performed on the Dresden

15  cluster at the MPI-C. N. Ojha and A. Pozzer thank Martin Körfer for his help with computing and data storage.



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





**Table 1.** RWP dataset used for studying the mixed layer height evolution.

| Month | Total no. of days | Clears sky days |
|---|---|---|
| November-2011 | 16 | 12 |
| December-2011 | 31 | 26 |
| January-2012 | 31 | 10 |
| February-2012 | 29 | 13 |
| March-2012 | 31 | 17 |



**Table 2.**Daytime (0500 -1000 UTC) monthly mean and percentage bias between RWP measurements and WRF simulations.

| Month | Observational mean (m) | Model mean (m) | Mean bias (m) | Percent bias (%) |
|---|---|---|---|---|
| November 2011 | 440±197 | 381±90 | -59 | -13.4 |
| December 2011 | 500±245 | 415±112 | -85 | -17 |
| January 2012 | 624±289 | 577±149 | -47 | -7.5 |
| February 2012 | 686±268 | 890±439 | +204 | +29.7 |
| March 2012 | 766±317 | 1350±664 | +584 | +76.2 |





**Table 3.** The initial and boundary conditions in the atmospheric boundary layer (ABL) and free troposphere (FT) as used in MXL/MESSy. All initial conditions are imposed at 03:00 UTC. t is the time elapsed since the start of the simulation (s) and td the length of the simulation (s). The subscripts s and e indicate values at the surface and the entrainment zone, respectively.

| Property | Value |
|---|---|
| Initial ABL height $h$ (m) | 280 |
| Subsidence rate $\omega$ (s$^{-1}$) | $2 \times 10^{-5}$ |
| Surface sensible heat flux $\overline{w'\theta'}_s$ (K m s$^{-1}$) | $0.24\sin(\pi t/t_d)$ |
| Entrainment/surface heat flux ratio $\beta = \overline{w'\theta'}_e / \overline{w'\theta'}_s$ (dimensionless) | 0.2 |
| Initial ABL potential temperature $\langle\theta\rangle$ (K) | 303.8 |
| Initial FT potential temperature $\langle\theta\rangle_{FT}$ (K) | 304.0 |
| Potential temperature lapse rate FT $\gamma_\theta$ (K m$^{-1}$) | 0.0065 |
| Surface latent heat flux $\overline{w'q'}_s$ (g kg$^{-1}$ m s$^{-1}$) | $0.11\sin(\pi t/t_d)$ |
| Initial ABL specific humidity $\langle q\rangle$ (g kg$^{-1}$) | 2.0 |
| Initial FT specific humidity $q_{FT}$ (g kg$^{-1}$) | 1.9 |
| Specific humidity lapse rate FT $q$ (g kg$^{-1}$ m$^{-1}$) | -0.0010 |



**Table 4.** Initial mixing ratio in ABL and FT, and surface emission fluxes of the reactants for MXL/MESSy runs. Species in the reaction mechanism that are not included in this table have an initial concentration of zero and no surface emissions. For O2 and N2 we have imposed the values $2 \times 10^8$ and $8 \times 10^8$ ppb, respectively.

| | $O_3$ | NO | $NO_2$ | BC | $CH_4$ | CO | $CO_2$ |
|---|---|---|---|---|---|---|---|
| Initial mixing ratio (ppb) | | | | (ng m$^{-3}$) | | | |
| ABL | 52 | 1.0 | 0.5 | 1300 | 1724 | 150 | 1300 |
| FT | 57 | 1.0 | 0.5 | 110 | 1724 | 150 | 110 |
| Surface emission flux | -0.55 ppb m/s | | | $5.5 \times 10^{-3}$ ng m$^{-2}$ h$^{-1}$ | | | |





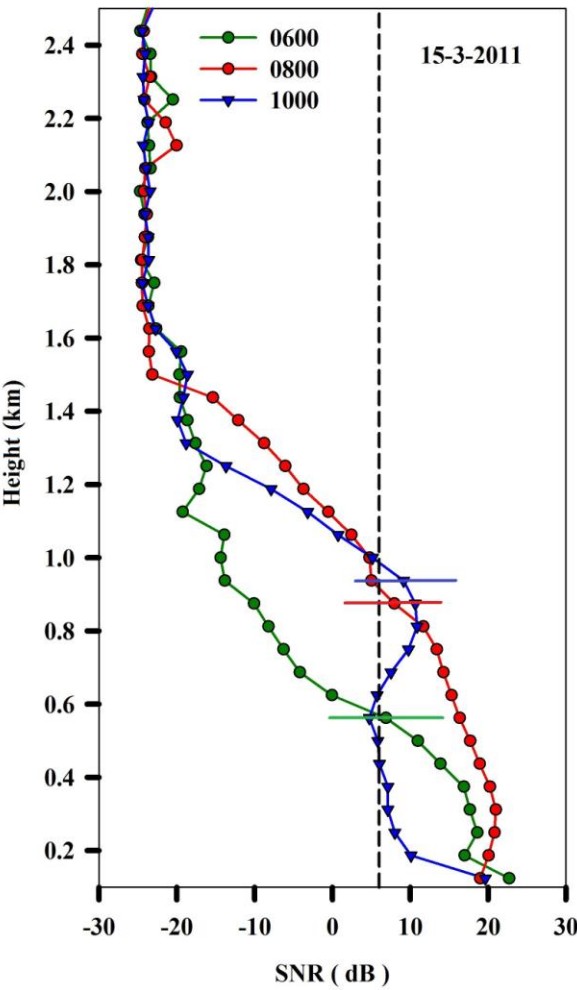

**Figure 1.** The signal-to-noise ratio profiles (15 minute averaged) on 15 March 2012 during the peak sunshine hours. The
5   estimated mixed layer height for each profile is marked by the horizontal bar.



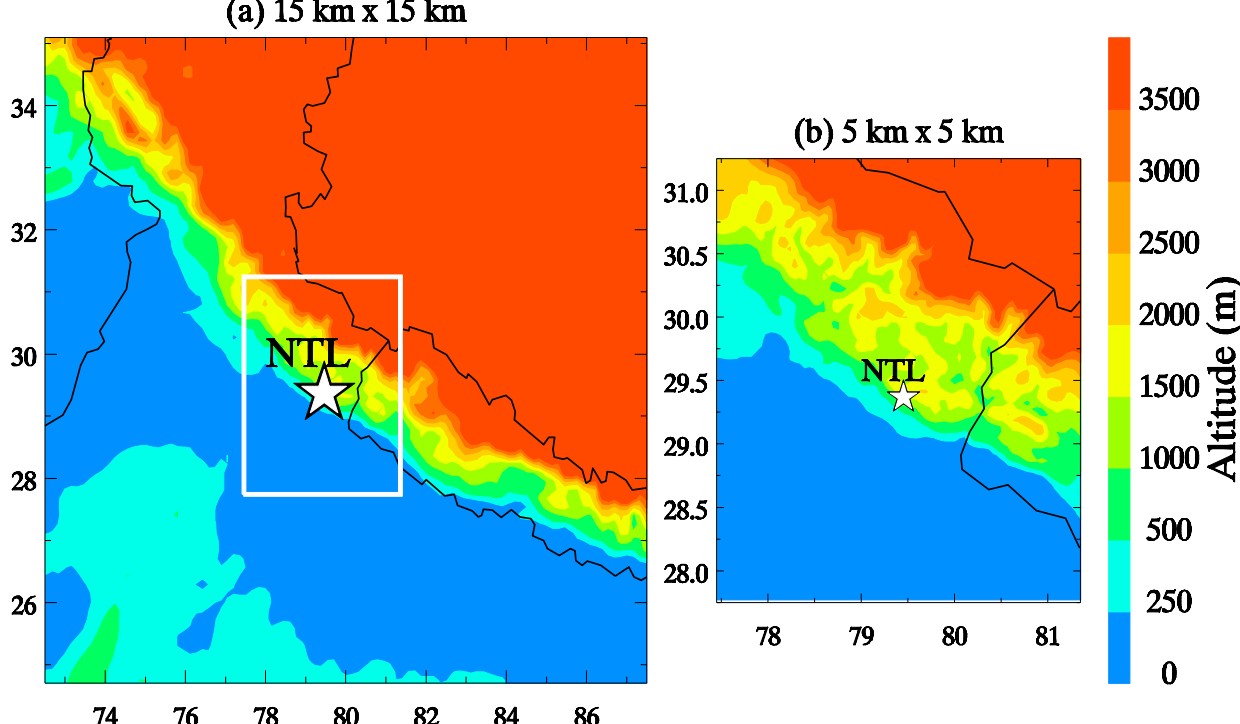

**Figure 2.** The WRF simulation domains used in the study are shown. The coarser domain (15km x 15 km) simulations are used to drive the simulation over the nested domain shown as white box in (a) and separately in (b). The geographical topography of the region and the location of measurement site Nainital (NTL) are also shown.





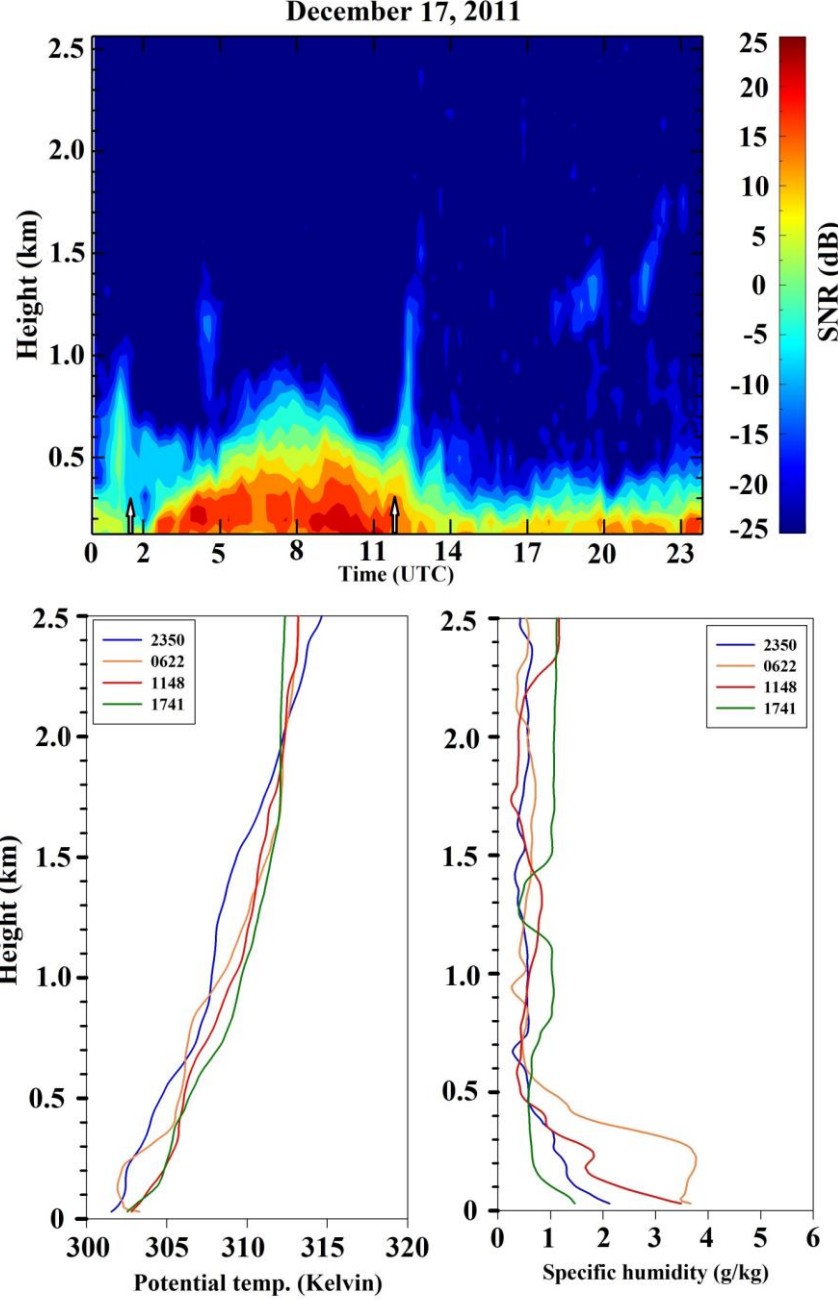

**Figure 3.** Mixed layer height for December 17, 2011. The upper panel shows range time intensity (RTI) plot of SNR (15 minute averaged) measured with RWP. The lower panel shows the vertical profiles of potential temperature and specific humidity derived from the four radiosonde flights conducted during the day (marked with different colors).





**Figure 4.** Same as Fig. 3 but for March 15, 2012**.**



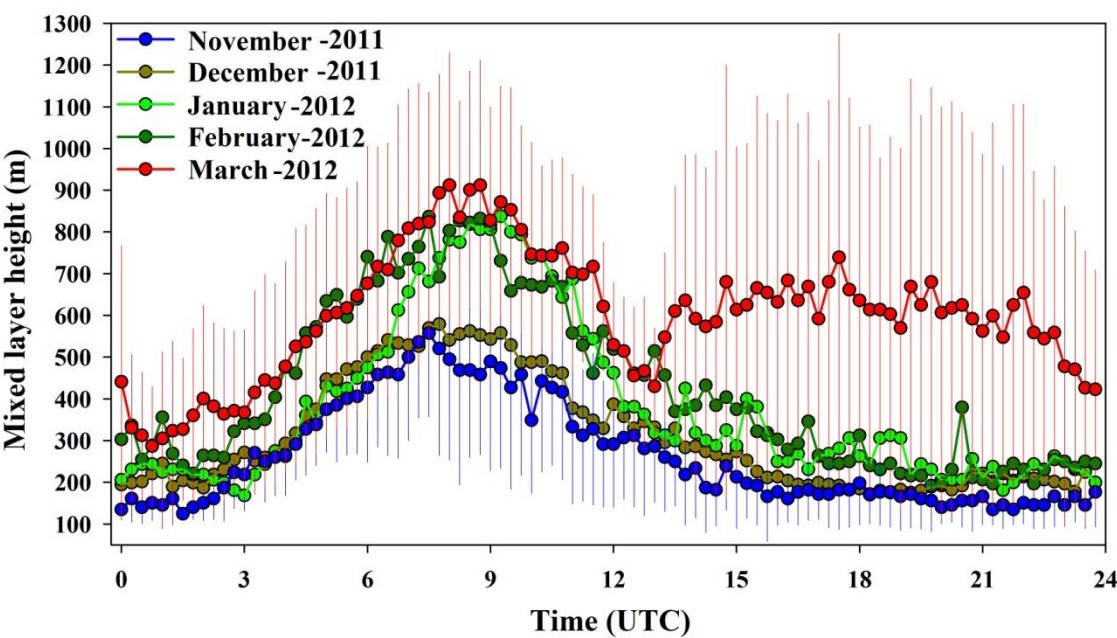

**Figure 5.** Monthly averaged mixed layer height determined from 15 minute averaged SNR profiles, measured with RWP.
For the sake of clarity, variability is only shown for November and March months.





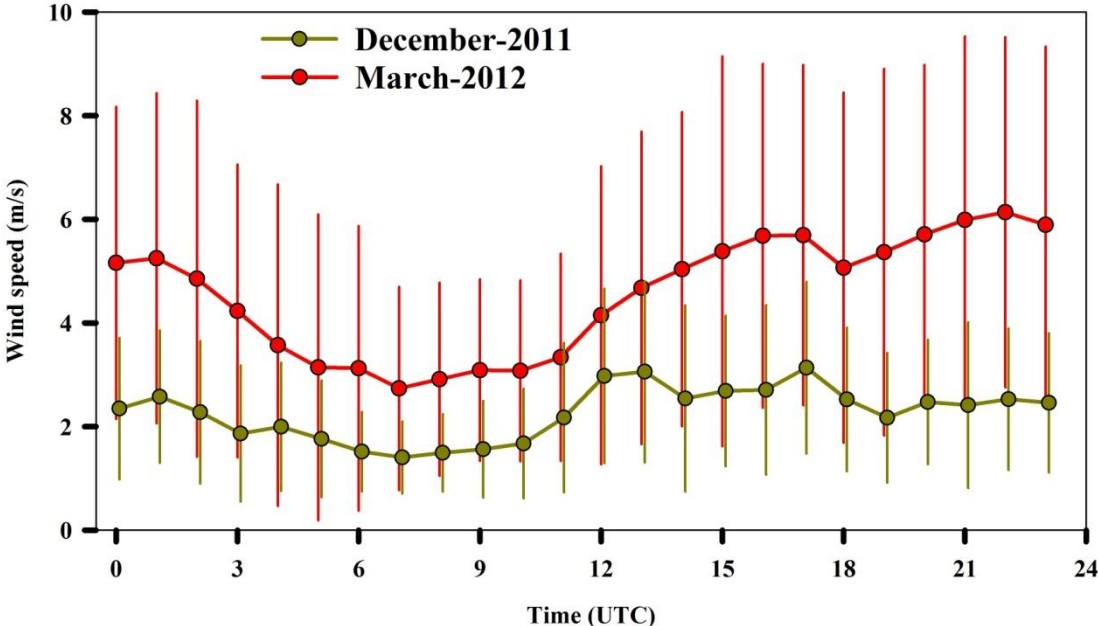

**Figure 6.** The monthly average diurnal variation in surface wind speed over the site during the clear sky days, selected for estimation of the mixed layer height as depicted in Fig. 3.





**Figure 7.** (a)A comparison of daytime (0500-1000 UTC) average boundary layer height from RWP measurements and WRF simulations at Nainital during the study period. Error bars represent 1-sigma standard deviation during daytime. (b) Correlation analysis between observational and model daytime boundary layer height.





**Figure 8.** Comparison of the monthly averaged diurnal variations in the boundary layer height determined from RWP measurements and WRF simulations at Nainital during the study period.







**Figure 9.** Diurnal variations in MXL simulated **(a)** boundary layer height, **(b)** surface ozone, and (c) Black Carbon (BC) on 15ᵗʰ March 2012 for four different simulations. The subs + adv represent simulation with subsidence and advection of cool air. Subs is with only subsidence, Adv is with only advection and ctrl is with neither subs. nor advection.


**Figure 10.** Sensitivity simulations of MXL simulated black carbon variations with different boundary layer dynamics. The blue line shows the control experiment, the green line a simulation with the initial potential temperature jump ($\Delta\theta_0$) set to 2 K, the red line a simulation with a free tropospheric potential temperature lapse rate ($\gamma_\theta$) set to 0.002 K/m and the turquoise line a simulation with $\gamma_\theta$ set to 0.009 K/m.