# Peer review of "Boundary layer evolution over the central Himalayas from Radio Wind Profiler and Model Simulations"

_Atmospheric Chemistry and Physics, 2016_

## Referee Comment (RC1) · Anonymous Referee #1 · 9 May 2016

General comments:

This paper presents 5 months of PBL height detection in case of sunny days by a radio wind profiler in the complex topography of the Himalayas. The measured PBL height is then compared to model results, and a sensitivity analysis of the PBL height, the ozone and BC concentrations depending on the model parameters is also done. PBL height measurements and modeling at high altitude sites in complex topography are rare and worth publishing. The authors explain quite well their methodology to detect the ML height from the SNR signal. Two examples are given that allow the reader to compare the ML heights estimated from WRP and from the virtual potential temperature and the specific humidity from radio-sounding measurements. If the ML height is clearly

reported on Fig. 3a and 4 a (RWP measurements), it should also be reported on Fig. 3b-c and 4b-c illustrating the ML height estimation from the virtual potential temperature and specific humidity. On the 17.12.2011, the specific humidity profiles give the ML height at about 0.3 km at 0622 and 1148, in good relation with the RWP. A ML height at about 0.25 km at 0622 is found from the virtual potential temperature (Parcel or bulk Richardson methods), but no ML height at 1148 (no unstable profile). As already mentioned in the quick review, a real validation of the MH height detection method by the RWP is necessary. For convective cases (unstable virtual potential temperature profiles), the Parcel or bulk Richardson should be applied. The stable boundary layer (stable potential temperature profiles) can be characterized by a surface-base temperature inversion and its top can be estimated by the height at which the gradient of the potential temperature ==0. The ML height can be also detected from the specific humidity gradient (maxima). A comparison of the three detection methods for the 78 good weather days is really necessary to assess the applied ML height detection method.

Minor comments:

- P. 1 line 12: "The standard criterion of the peak in the signal-to-noise ratio profile" : can you please shortly specify what is "the standard criterion".

- P. 1 line 15: is it really "the daytime average" or the average of the daily maximum of the boundary layer height ? Please specify if the BL height is a.g.l. (due to site elevation, it is probably not a.s.l.)

- p. 1 lines 20-21: revise the English (the introduction as a lot of language problem, please check it)

- P. 2 line 14: "to probe troposphere through atmospheric boundary layer": please rephrase.

- P.2 lines 19-21 and lines 24-26: rephrase, not comprehensible

- P. 2 line 27: distantly transported =Long-range transported?

[Figure]

- P. 2 line 28: " as well as . . .": rephrase

- P. 3 line 3: comprising four launches. . .

- P.3 lines 5-6: RWP wind measurements are used to provide continuous. . .

- P. 3 line 8 "the best possible" or " a better tool than " ?

- P.3 lines12-13: rephrase

- P.3 line 17: as simulated by a regional model: rephrase

- p.3 line 27: " as prominent as that as in high pressure regions": rephrase

- p. 4 2.1 observational site: please refer to Fig. 2 (and inverse therefore Fig 1 and 2)

- p. 4 line 14-15: rephrase

- p. 4 line 17 from a vertical beam or from one vertical beam ?

- p. 4 lines 25-30: please give a reference for the number given to describe the vertical structure of the PBL.

- p.4 line 30 to p. 5 line 6: please give a reference for the describe phenomena. At other high altitude site (for example at JFJ, see Ketterer et al., . . .) a maxima of the SNR was always measured in case of convective boundary layer. Perhaps the vertical velocities and the peak widths at the lowest time resolution (seconds or minutes, probably given as raw data) could help to describe this phenomena.

- Same place: as stipulate under "general comments", a validation of your detection method (< 6BD) has to be given.

- P.6 §3.1: please specify what is the method to diagnose the PBL height in your WRF model.

- P.7 line 6 can we really speak of a "smooth" diurnal cycle ?

- P.7 lines 10-12, Figures 3 and 4: the ML heights (calculated from both the plotted

profiles and from RWP) should be reported on the virtual potential temperature and humidity profiles. The ML heights estimated from the potential T and the specific humidity seems to be somewhat different on the 17.12.2011, whereas they seem to be in good accordance for both (0608 and 1139) sounding on the 15.3.2011. For that day, the ML heights detected from the RWP seem however much higher than the ones of the sounding.

- Case 15.3.2011: the virtual potential temperature and the specific humidity both show changes at 500 and 1800 m. The SNR signal is decreasing abruptly at about 0930. Can you explain these phenomena (for example also with vertical velocity if this is an effect of subsidence).

- P. 7 line 23:"both profiles" instead of "both the profiles"

- P.8 line 3: " with the least diurnal variability"

- P.8 lines 6-9 and §4.3: 1) What is the causes of the high wind velocities measured in March during night? Are they local valley winds, synoptic advective winds ? Does the wind direction change between day and night ? Are these winds well-known during spring in central Himalayas ? 2) the denomination of the boundary layer has to be clarified. LBL, ML, ABL, stable boundary layer and residual layer are all mentioned in §4.3. LBL induce a local effect and should be bounded with local winds induced by the topography. ABL has to be estimated by aerosol measurements (Lidar or ceilometers) or eventually by humidity profiles. Stable boundary layer should be estimated by the T profiles and the residual layer is often composed of several distinct layers. I hope that you can clarify this very interesting case with all the measurements at your disposal (wind horizontal and vertical velocities, wind direction, T and humidity profiles, perhaps wind measurements at other stations if synoptic advective winds are suspected or model results).

- P.8 line 16 "during the rest of the night"

[Figure]

- P.8 line 26 "in mixing depth. Errors. . ."

- Fig. 7b: if the points were colored as a function of the months, the deviations from the fitted slope could be estimated for each month.

- P. 9 line 14-15 and Table 2: Perhaps just mention if the monthly means of the model results are comprised in the variability ($\pm$ 1 sigma). The use of "more" is not appropriate

- P. 9 line 22: do you have an explanation or a suggestion to explain the overestimation of the model by a factor of 2 ?

- Fig. 9 and 10: The figure caption should describe what are the black and red points.

- Fig S1 and text related: it seems that the effect of the convective parameterization are quite small. I would not really conclude that the BL decrease is much faster between 11h and 13h. It seems that the high ML height at 13h is a kind of compensation of the low point at 12h.

- Fig. S3: the figure caption should be more comprehensive with the description of the points and the lines.

---

## Referee Comment (RC2) · Anonymous Referee #2 · 9 May 2016

Major question: the local (Himalayas Mountains) is a very dry. You can also see this looking to your profile of specific humidity. In this case, the SNR should be very weak, shouldn't? Could you comment this point? Minor questions: Page 1, Line 16: I think that the precision of the instrument did not permit an estimation of the BL with 1 decimal (for instance 439.6 m).Use the closest integer for the average (440 m) and the standart deviation (197 m). Page 2, line 10: I would suggest using potential temperature profile (not virtual potential temperature profile) as you are also using specific humidity. Page 3, line 2: use the word missing instead of lacking Page 3, line 10: contrasting periods: winter and spring? How contrasting are these seasons?! Do you have data at summer period (or stopped at March 2012)? Page 4, line 15: between 0.1 and 6 km.

[Figure]

Figure 1: the vertical scale is height (not altitude since there is 2000 m of amsl plus the vertical profile). Also, it is better to use March 15, 2012 as label for the date. What are the daytime hours?! What are the time zone differences for 06, 08 and 10 UTC? How have the authors computed the height of BL?! (described at page 5)

Page 5, lines 1-6: what are the relations between the description of the entrainment zone and the atypical SNR profile?! I did not completely understand the point raised at those lines. Could the authors clarify this point?! Page 5, line 7-8: what are the times of the radiosondes?! It was written 4 times per day, but I did not find the times (also some description of the radiosonde model used)? Clarify this point. Page 5, lines 10-12: what is the difference between LBL and PBL? Also, there is ML. All of them are refered to the boundary layer height. Clarify this point. Page 5, lines 24-25: FNL from GFS? Both are datasets used for WRF initialization, but they have different structures between them. Clarify this point.

Figure 2: I would suggest making a sign/arrow at sunrise and sunset at the top panel for better visualization of the convective/daytime conditions. Also, Kelvin should be written with capital letter. Why the profiles for virtual potential temperatures were shifted by 2 K? Explain this. The thetav and specific humidity profiles at 622 UTC shows clearly a shallow BL around 200 m. However, for the profile at 1148 (still daytime), the both profile has a stable pattern. Explain this.

Page 7, line 4: what are the differences (besides the months) for the chosen of these 2 contrasting days?! The authors should described them these difference as earlier as possible. Described in terms of sensible heat fluxes (or radiation energy budget values) Page 7, line 7: the profile at 1148 UTC should be convective/daytime instead of stable. Explain why this happen! Page 7, line 20-21: how a BL value around 800-1000 m (derived from the windprofiler) is consistent with an observations of 400-500 m derived from radiosondes?!Explain this point Page 8, Line 6: it is incorrect to say ML height at nighttime period. The authors should say nocturnal or stable boundary layer, not mixed layer (ML). Page 8, lines 20-21: " . . .ML decreases in depth, but this could

be attributed 20 to the rapid cooling of the surface". At this time (end of the night and near to the sunrise), there is no rapid cooling of the surface, as the NBL is very stable. Clarify this point. Page 8, line 27: It is missing a final point (between . . .mixing depth. Errors . . .) Page 9, lines 6-7: ".. Overall the model and observations are in reasonable agreement during the study period (r2 = 0.5)." Is 0.5 a reasonable agreement? What is the physical meaning of a negative value for intercept showed at Figure 7? Also, for Table 2, I suggest to use the integer values for height of BL! There are so many assumptions on the determination of the BL either by RWP or radiosondes that the integer values are a better representation of BL heights. Figure 3: upper panel: the authors should not draw a continuous line amongst the data as there are gaps (no data collected). See the example from end of December and beginning of March. Lower panel: the both scales should be the same, plot the line 1:1 and explain the physical meaning of a negative value for intercept.

Page 10, line 4: potential temperature?! It was used virtual potential temperature for the entire text. I agree to use potential temperature (instead of virtual potential temperature), but it should be along the entire document. Page 12, Lines 9-12: it is very strong the sentence saying that RWP gives the best temporal estimates. If you have a ceilometer and/or a microwave radiometer, you also have very good estimates of the BL heights. By the other hand, these instruments (ceilometer and radiometer) are much more simple to use thank a wind profiler. So, I suggest to re-write this sentence. Page 12: Line 29: torrential downpour. Can it be replaced by torrential rain? The line below, I think that it is year long (not yearlong).

---

## Author Comment (AC1) · 8 Jul 2016

Please find the response to reviewer's comments and revised manuscript with changes highlighted in the zip file attached as the supplement.

Please also note the supplement to this comment: http://www.atmos-chem-phys-discuss.net/acp-2016-101/acp-2016-101-AC1-supplement.zip

---

## Author Response (AR1)

**Response to the comments of reviewer #1**

**Authors are thankful to the reviewer for his/her careful evaluation of the manuscript. We appreciate the reviewer's suggestion to thoroughly validate the criterion for ML height detection and testing of various methods to determine the ML height from radiosonde. The inclusion of recent and most relevant reference (Ketterer et al., 2014) was of paramount importance in explaining the RWP measurements over a complex terrain as suggested by the referee. Reviewer's valuable comments have helped us to greatly improve our manuscript. We have carefully gone through the comments and implemented accordingly. Reviewer's comments are in regular font and our replies are in bold font characters.**

**General comments:**
This paper presents 5 months of PBL height detection in case of sunny days by a radio wind profiler in the complex topography of the Himalayas. The measured PBL height is then compared to model results, and a sensitivity analysis of the PBL height, the ozone and BC concentrations depending on the model parameters is also done. PBL height measurements and modeling at high altitude sites in complex topography are rare and worth publishing.

The authors explain quite well their methodology to detect the ML height from the SNR signal. Two examples are given that allow the reader to compare the ML heights estimated from WRP and from the virtual potential temperature and the specific humidity from radio-sounding measurements. If the ML height is clearly reported on Fig. 3a and 4 a (RWP measurements), it should also be reported on Fig. 3b-c and 4b-c illustrating the ML height estimation from the virtual potential temperature and specific humidity. On the 17.12.2011, the specific humidity profiles give the ML height at about 0.3 km at 0622 and 1148, in good relation with the RWP. A ML height at about 0.25 km at 0622 is found from the virtual potential temperature (Parcel or bulk Richardson methods), but no ML height at 1148 (no unstable profile).
As already mentioned in the quick review, a real validation of the MH height detection method by the RWP is necessary. For convective cases (unstable virtual potential temperature profiles), the Parcel or bulk Richardson should be applied. The stable boundary layer (stable potential temperature profiles) can be characterized by a surface-base temperature inversion and its top can be estimated by the height at which the gradient of the potential temperature ==0. The ML height can be also detected from the specific humidity gradient (maxima). A comparison of the three detection methods for the 78 good weather days is really necessary to assess the applied ML height detection method.

**Response:**
**It is well understood that the mixed layer (ML) height/local boundary layer over complex terrain is generally not as prominent and well-evolved as that over the flat terrain, due to the number of factors affecting the state of the atmosphere, such as sudden dry/moist air**

advection within 1 km above mountain top, the strengthening of exchange processes and orographic influences. Therefore, it becomes difficult to obtain typical features of mixed layer height over such a complex terrain. We have tested the bulk Richardson, specific humidity gradient (maxima), and virtual potential temperature profile gradient method (now used potential temperature only, see response to reviewer #2) as suggested. But, expectedly over a complex terrain, only the specific humidity gradient (maxima) method provided the promising results and has been presented in the manuscript.

We have carried out a detailed correlation analysis for ML derived from specific humidity gradient method vs RWP (SNR method), and the maximum correlation ($r^2 = 0.70$) between the two is found to be at the SNR value of 6 dB for the noon-time (0600 UTC) and evening (1200 UTC) profiles that closely correspond to the convective state of boundary layer over the site.

Nevertheless, we have discarded the cases of minimum ML height estimation from RWP (RWP starts measurements beyond 124 m in the vertical) and radiosonde, as well as the days of very weak ($< 10$ g kg$^{-1}$ km$^{-1}$) gradient in specific humidity profiles. We have considered only the day time cases, as the agreement for midnight and early morning profiles (1800 and 0000 UTC) between ML derived from radiosonde profiles and RWP profiles turns out to be very poor for all of the known methods. This may be due to the orographic effects which are dominant in nighttime and do not represent convective ML. The disagreement may also be attributed to the topography of the site (a mountain peak) and the drift in the radiosonde, which thus provides measurements of ML height over the adjoining valleys or ridges at nighttime. Since the ML top follows the topography over mountainous terrain, and the degree to which ML top follows the topography is minimum at noontime and afternoon hours (De Wekker and Kossmann, 2015). Hence, a higher correlation between radiosonde ML height estimation and RWP derived LBL height is certainly expected to be significant only during the noontime to evening hours. The correlation analysis for ML derived from RWP vs radiosonde is given in the Figure C1 below:

[Figure]

**Figure C1: Correlation analysis for RWP (ML) vs Radiosonde (ML)**

**Minor comments:**

- P. 1 line 12: "The standard criterion of the peak in the signal-to-noise ratio profile": can you please shortly specify what is "the standard criterion".

**Response: The standard criterion is the peak detection in the SNR profile that corresponds to entrainment zone or top of the mixed layer or inversion (representing the Convective Boundary Layer top) and is generally the standard method of mixed layer height determination with RWP (Wyngaard and LeMone, 1980; Fairall 1991; Angevine 1994 and Simpson et al., 2007).**

- P. 1 line 15: is it really "the daytime average" or the average of the daily maximum of the boundary layer height ? Please specify if the BL height is a.g.l. (due to site elevation, it is probably not a.s.l.).

**Response: Yes, it is the daytime average (0500 to 1000 UTC) averaged boundary layer height. The BL height has now been specified in AGL in the revised manuscript.**

- p. 1 lines 20-21: revise the English (the introduction as a lot of language problem, please check it).

**Response: The mentioned lines are corrected. English in the 'Introduction' section is revised and corrected to the possible extent.**

- P. 2 line 14: "to probe troposphere through atmospheric boundary layer": please rephrase.

**Response: The sentence has been rephrased.**

- P.2 lines 19-21 and lines 24-26: rephrase, not comprehensible

**Response: The sentences have been rephrased in the revised manuscript.**

- P. 2 line 27: distantly transported =Long-range transported?
- P. 2 line 28: " as well as : : :": rephrase

**Response: The entire sentence has been rephrased for clarity.**

- P. 3 line 3: comprising four launches: : :

**Response: Correction incorporated.**

- P.3 lines 5-6: RWP wind measurements are used to provide continuous: : :

**Response: The sentence has been rephrased for clarity.**

- P. 3 line 8 "the best possible" or " a better tool than " ?

**Response: For the site under consideration, the use of word best seems to be appropriate.**

- P.3 lines12-13: rephrase

**Response: The entire sentence has been rephrased for clarity.**

- P.3 line 17: as simulated by a regional model: rephrase

**Response: The sentence has been rephrased.**

- p.3 line 27: " as prominent as that as in high pressure regions": rephrase

**Response: The sentence has been rephrased.**

- p. 4 2.1 observational site: please refer to Fig. 2 (and inverse therefore Fig 1 and 2).

**Response: Figure 2 is in-line and associated with section 3.1, and basically describes the model domain but not the site topography. Therefore, we feel it should be retained in the same order to maintain the continuity.**

- p. 4 line 14-15: rephrase
**Response: Implemented.**

- p. 4 line 17 from a vertical beam or from one vertical beam?
**Response: Correction has been incorporated.**

- p. 4 lines 25-30: please give a reference for the number given to describe the vertical structure of the PBL.
**Response: The reference of Angevine et al., 1994 has now been mentioned at suitable place in the revised manuscript.**

- p.4 line 30 to p. 5 line 6: please give a reference for the describe phenomena. At other high altitude site (for example at JFJ, see Ketterer et al., : : :) a maxima of the SNR was always measured in case of convective boundary layer. Perhaps the vertical velocities and the peak widths at the lowest time resolution (seconds or minutes, probably given as raw data) could help to describe this phenomena.
**Response: The reference to Ketterer et al., 2014 is provided in the revised manuscript along with a brief discussion (Page: 5, Line: 4-6). We agree with the referee's view of analyzing raw data for vertical velocities and peak width at available 30 s resolution and the details are presented subsequently, however, here we focused our analysis only for SNR method. Additionally, the vertical velocities and peak widths are the first and second moments estimated from the SNR itself. Considering that the evolution of boundary layer is a process that responds to surface forcings and takes place in an hour or so, we have averaged the data presented here, for 15 minutes which still is a better resolution for a mixing depth evolution over a mountain top. Figure C2 provided here gives a picture of variations in vertical velocity.**
**The Wind profiler measurements presented in Ketterer et al., 2014 although being made over complex mountainous terrain (installed at Kleine Scheidegg) revealed a peak in the SNR profile during convective daytime conditions. It may however be noted that the station at Kleine Scheidegg is situated on a mountain pass, whereas the measurements presented in this study were taken over a mountain peak (with no obstruction to winds from any direction); hence difference in characteristics of LBL could be anticipated due to differences in the topography.**

-Same place: as stipulate under "general comments", a validation of your detection method (< 6BD) has to be given.

**Response: The validation of the ML detection method has now been provided in the revised manuscript.**

- P.6 §3.1: please specify what is the method to diagnose the PBL height in your WRF model.

**Response: WRF simulations presented here used the Mellor Yamada Janjic (MYJ) scheme, which employs the TKE method to diagnose the PBL height (Janjic, 2002; Mellor and Yamada, 1982) (Page: 6, Lines: 24-26).**

- P.7 line 6 can we really speak of a "smooth" diurnal cycle ?

**Response: Yes, for December 17 2011, we can certainly speak of a "smooth" diurnal cycle; since the fluctuations in ML height (over the duration of 15 minutes) are minimum that is equivalent to the vertical resolution of the RWP observations.**

- P.7 lines 10-12, Figures 3 and 4: the ML heights (calculated from both the plotted profiles and from RWP) should be reported on the virtual potential temperature and humidity profiles. The ML heights estimated from the potential T and the specific humidity seems to be somewhat different on the 17.12.2011, whereas they seem to be in good accordance for both (0608 and 1139) sounding on the 15.3.2011. For that day, the ML heights detected from the RWP seem however much higher than the ones of the sounding.

**Response: The ML height estimated from the specific humidity gradient (maxima) method has now been presented in the Figures 3 and 4. From the gradient in specific humidity (maxima) the ML height on December 17 2011 is estimated to be 340 m and 300 m for the 0622 and 1148 UTC profile, whereas from the RWP (SNR 6 dB criterion) the ML height is 437 and 374 m respectively. Similarly for March 15, 2012 the ML height comes out to be 500 and 540 m for the 0608 and 1139 UTC radiosonde profile, whereas from the RWP the ML height is estimated to be 562 and 750 m respectively.**

- Case 15.3.2011: the virtual potential temperature and the specific humidity both show changes at 500 and 1800 m. The SNR signal is decreasing abruptly at about 0930. Can you explain these phenomena (for example also with vertical velocity if this is an effect of subsidence).

**Response: There may be multiple inversions in the atmosphere above the complex topography since such features are commonly anticipated. Possible causes may be the cold/dry air advection in the transition zones of wind field e. g. between 4 and 6 km above mean sea level and the interactions of tropical air masses with that of the mid-latitude. Moreover, the inversions at 500m are distinct as compared to that at 1800m, and such a high mixing depth over mountain topography during the evening hours is generally not expected. Figure C2 provides the vertical velocity picture on the day, where the subsidence can be seen in the atmosphere up to about 2km in the afternoon hours. Though the SNR is found to decrease but the strength of the signal still remains above our detection limit of 6 dB.**

**March 15, 2012**

[Figure]

**Figure C2: vertical velocity contour plot on 15$^{th}$ March 2012.**

- P. 7 line 23:"both profiles" instead of "both the profiles"
**Response: Correction incorporated.**

- P.8 line 3: " with the least diurnal variability"
**Response: Correction implemented.**

- P.8 lines 6-9 and §4.3: 1) What is the causes of the high wind velocities measured in March during night? Are they local valley winds, synoptic advective winds? Does the wind direction change between day and night? Are these winds well-known during spring in central Himalayas?
**Response: During March (first month of the spring season), the site experiences strong synoptic north-westerly winds, and as a result of interaction between mountain wind system and synoptic winds, a small but distinct change in wind direction is observed with the flow and being northwesterly during 1400 to 0700 UTC (i.e., nighttime and forenoon hours) and westerly winds during afternoon hours (0800 to 1100 UTC). These variations in wind speed and direction are well known characteristics of meteorological conditions in the central Himalayas during spring season (Solanki et al., 2016).**
2) the denomination of the boundary layer has to be clarified. LBL, ML, ABL, stable boundary layer and residual layer are all mentioned in §4.3. LBL induce a local effect and should be

bounded with local winds induced by the topography. ABL has to be estimated by aerosol measurements (Lidar or ceilometers) or eventually by humidity profiles. Stable boundary layer should be estimated by the T profiles and the residual layer is often composed of several distinct layers. I hope that you can clarify this very interesting case with all the measurements at your disposal (wind horizontal and vertical velocities, wind direction, T and humidity profiles, perhaps wind measurements at other stations if synoptic advective winds are suspected or model results).

**Response: The section 4.3 has been rewritten for clarity and the results presented in section 4.3 are in agreements with the conclusions of Solanki et al. (2016), a study on surface layer characteristics over the site during the spring season. We agree with the referee for his suggestion, but as seen in the case of spring, the nocturnal local boundary layer can evolve as a result of orographic lifting particularly under high wind conditions ($>5$ ms$^{-1}$). A detailed study only on nocturnal boundary layer may be carried out separately along with the utilization of available co-located measurements. However, a general picture on synoptic conditions and surface winds has been already presented in other studies carried out at this site earlier (Kumar et al. 2010).**

- P.8 line 16 "during the rest of the night"
**Response: Correction incorporated.**

- P.8 line 26 "in mixing depth. Errors…"
**Response: Correction incorporated.**

- Fig. 7b: if the points were colored as a function of the months, the deviations from the fitted slope could be estimated for each month.
**Response: Thanks for the suggestion. We tried to calculate statistics for all months individually, however, some of the months have very less number of data counts / clear days (Table 1) and more scatter in particular during January. Therefore, we segregate data in three seasons as shown below to find $r^2$ values ranging from 0.3 to 0.7, slope (m) from 0.5 to 2 and intercept (c) from -316 to 180. We have maximum number of data counts in November and hence the highest value (0.7) of correlation.**

[Figure]

**Fig C3: Season-wise scatter plot analysis between observational and modeled daytime mixed layer height. A 1:1 line is also shown.**

- P. 9 line 14-15 and Table 2: Perhaps just mention if the monthly means of the model results are comprised in the variability (± 1 sigma). The use of "more" is not appropriate
**Response: Suggestion incorporated.**

- P. 9 line 22: do you have an explanation or a suggestion to explain the overestimation of the model by a factor of 2?
**Response: The overestimation of mixed layer could be associated with the parameterization of boundary layer and the land-surface processes, in addition to the errors in simulations of other meteorological variables, and effects of unresolved topographical features. This is discussed in the revised manuscript (Page: 10, Lines: 10-13).**

- Fig. 9 and 10: The figure caption should describe what are the black and red points.
**Response: Suggestion incorporated.**

- Fig S1 and text related: it seems that the effect of the convective parameterization are quite small. I would not really conclude that the BL decrease is much faster between 11h and 13h. It seems that the high ML height at 13h is a kind of compensation of the low point at 12h.

**Response: Turning off the convective parameterization does lead to relatively smoother settling in the evening and the pointed out sentence is revised accordingly (Supplementary material-caption of Figure S1).**

- Fig. S3: the figure caption should be more comprehensive with the description of the points and the lines.
**Response: Suggestion incorporated.**

**References used in the response to referee 1:**

Angevine, W., White, A., Avery, S.: Boundary-layer depth and entrainment zone characterization with a boundary-layer profiler. Bound.-Lay. Meteorol., 68, 375–385, 1994.

De Wekker, S.F.J., and Kossmann, M.: Convective Boundary Layer Heights Over Mountainous Terrain—A Review of Concepts. Front. Earth. Sci., 3, 77, doi: 10.3389/feart.2015.00077, 2015.

Fairall, C.W.: The humidity/temperature sensitivity of clear-air radars for the cloud free convective boundary layer. Journal of Applied Meteorology, 8, 1064–1074, 1991.

Janjic, Z.I.: Nonsingular Implementation of the Mellor-Yamada Level 2.5 Scheme in the NCEP Meso model. NCEP Office Note, 437, 61 pp, 2002.

Ketterer, C., Zieger, P., Bukowiecki, N., Collaud Coen, M., Maier, O., Ruffieux, D., and Weingartner, E.: Investigation of the planetary boundary layer in the Swiss Alps using remote sensing and in-situ measurements, Bound.-Lay. Meteorol., 151, 317-334, doi:10.1007/s10546-013-9897-8, 2014.

Kumar, R., Naja, M., Venkataramani, S., and Wild, O.: Variations in surface ozone at Nainital, a high altitude site in the Central Himalayas. *J. Geophys. Res.*, 115, D16302, doi: 10.1029/2009JD013715, 2010.

Mellor, G.L., and Yamada, T.: Development of a turbulence closure model for geophysical fluid problems, Rev. Geophys., 20(4), 851–875, doi:10.1029/RG020i004p00851, 1982.

Simpson, M., Raman, S., Lundquist, J.K., Leach, M.: A study of the variation of urban mixed layer heights. Atmos. Environ., 41, 6923-6930, 2007.

Solanki, R., Singh, N., Kiran Kumar, N.V.P., Rajeev, K., and Dhaka, S.K.: Time variability of surface-layer characteristics over a mountain ridge in the central Himalayas during the spring season. Bound.-Lay. Meteorol., 158, 453-471, doi: 10.1007/s10546-015-0098-5, 2016.

Wyngaard, J.C., LeMone, M.A.: Behavior of the refractive index structure parameter in the entraining convective boundary layer. Journal of Atmospheric Science 37, 1573–1585, 1980.

**Response to the comments of reviewer #2.**

**Authors thank to the referee for the careful evaluation of the manuscript, providing the valuable suggestions and comments that helped us to improve the manuscript significantly. We have cautiously gone through the comments and implemented accordingly. Reviewer's comments are in regular font and our replies are in bold font characters**.

**Major question**: the local (Himalayas Mountains) is a very dry. You can also see this looking to your profile of specific humidity. In this case, the SNR should be very weak, shouldn´t? Could you comment this point?

**Response**: **It is worth mentioning here that wind profiler basically detects the subtle fluctuations in the radio refractive index gradients, which in turn depend upon temperature and humidity gradients. Humidity gradients in the few hundred meters above surface will always be dominating due to orographic effects. We agree that SNR will not be that strong particularly in the absence of solar radiation, but as a result of mixing due to orographic influences and slope winds the SNR will always be positive (in our case) in the nocturnal local boundary layer over a mountain peak.**

**Minor questions**: Page 1, Line 16: I think that the precision of the instrument did not permit an estimation of the BL with 1 decimal (for instance 439.6 m).Use the closest integer for the average (440 m) and the standard deviation (197 m).

**Response**: **We agree with your suggestion, and the BL height values have now been reported to the closest integer in meters only.**

Page 2, line 10: I would suggest using potential temperature profile (not virtual potential temperature profile) as you are also using specific humidity.

**Response**: **Thank you for your suggestion, the potential temperature profiles have been presented in the revised manuscript throughout.**

Page 3, line 2: use the word missing instead of lacking.

**Response**: **Correction incorporated as per suggestion.**

Page 3, line 10: contrasting periods: winter and spring? How contrasting are these seasons?! Do you have data at summer period (or stopped at March 2012)?

**Response**: **The contrasting periods, referring to winter and spring (March only) have seasonal averaged sensible heat flux (SHF) of 50 Wm$^{-2}$ and 17 Wm$^{-2}$ respectively i.e. the SHF in spring is almost three times of that in the winter (Solanki et al., 2016).**

Page 4, line 15: between 0.1 and 6 km.

**Response**: **Correction incorporated.**

Figure 1: the vertical scale is height (not altitude since there is 2000 m of amsl plus the vertical profile). Also, it is better to use March 15, 2012 as label for the date. What are the daytime

hours?! What are the time zone differences for 06, 08 and 10 UTC? How have the authors computed the height of BL?! (described at page 5)

**Response**: **Altitude changed to height throughout the manuscript. Date format has also been changed. For the site, local time is UTC +5.5 Hours (i.e. 5 hours and 30 minutes ahead of UTC), and this is also mentioned on page 7 in line 8. We have quoted about BL height on page 5, this height has only been speculated from the RWP and radiosonde profiles.**

Page 5, lines 1-6: what are the relations between the description of the entrainment zone and the a typical SNR profile?! I did not completely understand the point raised at those lines. Could the authors clarify this point?!

**Response**: **We have speculated about the characteristics of entrainment zone over complex mountainous terrain in these lines. It has been stated that, over flat homogeneous terrain, a peak in the SNR profile implies the entrainment zone or inversion layer; however such a feature was non-existent in the profiles measured at the site under consideration.**

Page 5, line 7-8: what are the times of the radiosondes?! It was written 4 times per day, but I did not find the times (also some description of the radiosonde model used)? Clarify this point.

**Response**: **The general times of radiosonde launches were (0600, 1200, 1800 and 0000 UTC), however the exact launching times of radiosonde ascents illustrated in the plots have been mention in the figures itself. The radiosonde model being used ("Vaisala RS92-SGP"), has now been mentioned in the revised manuscript.**

Page 5, lines 10-12: what is the difference between LBL and PBL? Also, there is ML. All of them are refered to the boundary layer height. Clarify this point.

**Response**: **LBL basically refers to the inhomogeneity in the PBL over inhomogeneous terrain; these inhomogeneities can arise as a result of variation in topography, moisture content, etc. Through boundary layer height we refer to the height to which the influence of the underlying surface is discernible, being referred to as ML during daytime and stable boundary layer during nighttime. More precisely, the extent of mixing in the day time as a result of solar heating is termed as ML which may be called PBL and LBL for flat and complex terrain respectively.**

Page 5, lines 24-25: FNL from GFS? Both are datasets used for WRF initialization, but they have different structures between them. Clarify this point.

**Response**: **Description of NCEP data is provided in the manuscript.**

Figure 2: I would suggest making a sign/arrow at sunrise and sunset at the top panel for better visualization of the convective/daytime conditions. Also, Kelvin should be written with capital letter. Why the profiles for virtual potential temperatures were shifted by 2 K? Explain this.

**Response**: **Arrow has now been drawn in the upper panel (RTI plots) of figure 3 and 4 marking the sunrise and sunset times. 'Kelvin' is now written with capital 'K' in the revised manuscript and figures. The reason behind shifting virtual potential Temperature Profiles by 2K was to make the profiles in the figure distinct; however, in the new figures of potential temperature profiles, this shifting has been removed.**

The thetav and specific humidity profiles at 622 UTC shows clearly a shallow BL around 200 m. However, for the profile at 1148 (still daytime), the both profile has a stable pattern. Explain this.
**Response**: **Yes, for 0622 UTC profile that is indeed the case, however, on December 17, 2011, the sunset time is 11:47 UTC. Thus the 1148 UTC profile reflects post sunset (dusk) conditions over site, wherein a rapid decrease in the moisture content is observed, with a minor bump in the profile at 200 m. Hence, it can be speculated that the downslope flows just triggering before sunset (due to rapid cooling of the mountain peak) might result in such a transition in humidity profiles. These downslope flows grow in depth after sunset that probably reach up to 200 m above surface with weaker magnitudes and results in an overall reduction with a minor bump at 200 m, as seen in the humidity profile.**

Page 7, line 4: what are the differences (besides the months) for the chosen of these 2 contrasting days?! The authors should described them these difference as earlier as possible. Described in terms of sensible heat fluxes (or radiation energy budget values)
**Response**: **The contrasting days have been taken from spring and winter season for which the seasonal averaged sensible heat flux is 50 Wm$^{-2}$ and 17 Wm$^{-2}$ respectively, same has been clarified in the revised manuscript.**

Page 7, line 7: the profile at 1148 UTC should be convective / daytime instead of stable. Explain why this happen!
**Response**: **The 1148 UTC is not convective daytime, since the sunset time is 11:47 UTC on December 17, 2011, and due to rapid cooling of the ridges and mountain peaks during sunset, the profile represents a stable atmosphere over the site.**

Page 7, line 20-21: how a BL value around 800-1000 m (derived from the wind profiler) is consistent with an observations of 400-500 m derived from radiosondes?!Explain this point
**Response**: **From the RTI plot and 6 dB criterion the BL height value is found to be 562 m at 0608 UTC and 750 m at 1139 UTC, which is nearly consistent (considering the resolution of 62.6 m for RWP measurements) with the radiosonde inversion height of 500 and 540 m respectively.**

Page 8, Line 6: it is incorrect to say ML height at nighttime period. The authors should say nocturnal or stable boundary layer, not mixed layer (ML).
**Response**: **We agree with the reviewer, and the term nocturnal boundary layer is now being used in the manuscript for nighttime period.**

Page 8, lines 20-21: " …ML decreases in depth, but this could be attributed 20 to the rapid cooling of the surface". At this time (end of the night and near to the sunrise), there is no rapid cooling of the surface, as the NBL is very stable. Clarify this point.
**Response**: **The section has been rephrased along with the inclusion of one important reference in the revised manuscript.**

Page 8, line 27: It is missing a final point (between …mixing depth. Errors …).
**Response**: **Thank you for pointing out the mistake. The sentences have been re-written for clarity.**

Page 9, lines 6-7: ".. Overall the model and observations are in reasonable agreement during the study period (r2 = 0.5)." Is 0.5 a reasonable agreement? What is the physical meaning of a negative value for intercept showed at Figure 7?

**Response**: **We agree that $r^2$ values of 0.5 do not represent, in general, a very good agreement; however, over the complex terrains such as Himalayas, models generally fail to resolve the topography and reproduce the meteorology affected by local winds and convection. Therefore, in this specific context of model performance over Himalayas, we said that $r^2$ value of 0.5 is reasonable. Intercept basically represents the difference in the detection limit of RWP measurements and model simulations, negative value might be due to the highly complex topography around the peak.**

Also, for Table 2, I suggest to use the integer values for height of BL! There are so many assumptions on the determination of the BL either by RWP or radiosondes that the integer values are a better representation of BL heights.

**Response**: **Correction incorporated, integer values are being used for BL height throughout the revised manuscript.**

Figure 3: upper panel: the authors should not draw a continuous line amongst the data as there are gaps (no data collected). See the example from end of December and beginning of March.

**Response**: **The line has been removed. However, there is no data gap but a 15 min mean is taken and a little lower SNR than 6dB in the nocturnal measurements at few occasions.**

Lower panel: the both scales should be the same, plot the line 1:1 and explain the physical meaning of a negative value for intercept.

**Response**: **Both scales have been made same now and 1:1 line has also been added. Negative Intercept is explained in one of the comments above, however, this is also perceptible that model is resolving the local boundary from the valley and not from the top in the night time.**

Page 10, line 4: potential temperature?! It was used virtual potential temperature for the entire text. I agree to use potential temperature (instead of virtual potential temperature), but it should be along the entire document.

**Response**: **We agree with the reviewer, and in the revised manuscript potential temperature has been used instead of virtual potential temperature.**

Page 12, Lines 9-12: it is very strong the sentence saying that RWP gives the best temporal estimates. If you have a ceilometer and/or a microwave radiometer, you also have very good estimates of the BL heights. By the other hand, these instruments (ceilometer and radiometer) are much more simple to use thank a wind profiler. So, I suggest to re-write this sentence.

**Response**: **The sentence has been re-written as suggested.**

Page 12: Line 29: torrential downpour. Can it be replaced by torrential rain? The line below, I think that it is year long (not yearlong).

**Response**: **Correction incorporated.**

**References used in the response to referee 2:**

[revised manuscript text omitted]